# Reasoning in Reasoning

*A Hierarchical Framework for Neural Theorem Proving*

**Ziyu Ye**[*,1]**, Jiacheng Chen**[2]**, Jonathan Light**[3]**, Yifei Wang**[4]**, Jiankai Sun**[5]**, Mac Schwager**[5]**,
**Philip Torr**[6,7]**, Guohao Li**[7,8]**, Yuxin Chen**[1]**, Kaiyu Yang**[9]**, Yisong Yue**[2]**, Ziniu Hu**[2]

[1]The University of Chicago, [2]California Institute of Technology, [3]RPI, [4]MIT CSAIL,
[5]Stanford University, [6]University of Oxford, [7]Eigent AI, [8]Camel-AI.org, [9]Meta FAIR
Code: github.com/ziyu-deep/reasoning-in-reasoning

## Abstract

Learning to do complex reasoning is the central objective of artificial intelligence. Autoregressive language models have shown promise in generating intermediate steps for problem solving; however, complex reasoning tasks such as theorem proving still present challenges due to the vast search spaces. Classical works have considered reasoning by searching, *e.g.*, expanding the reasoning space with tree search to explore intermediate steps; and reasoning by decomposing, *i.e.*, breaking down the problem into higher-level thoughts that prompt lower-level actions. In this work, we develop Reasoning in Reasoning (RiR), a hierarchical framework that formally unifies *decomposing* and *search* by a planner-actor game. Using neural theorem proving as a representative task, our approach breaks down complex theorem proving problems into achievable sub-goals for abstraction over formal proofsteps, giving models: (i) improved generalizability for reasoning step generation, (ii) a more compact and informative search space for reasoning trajectories, and (iii) an efficient mechanism for learning to plan. We empirically show that RiR achieves concrete performance gain on popular theorem proving datasets including LeanDojo and miniF2F while being highly efficient (*e.g.*, RiR is nearly 3x faster over the existing state-of-the-art baseline on miniF2F). We further present information-theoretic conjectures on the principles driving RiR's effectiveness.

*A very powerful approach is to attempt to eliminate everything from the problem except the essentials; that is, cut it down to size. Very often, if you can solve the simple problem, you can add refinements to the solution of this, until you get back to the solution of the one you started with.*

*– Claude Shannon*

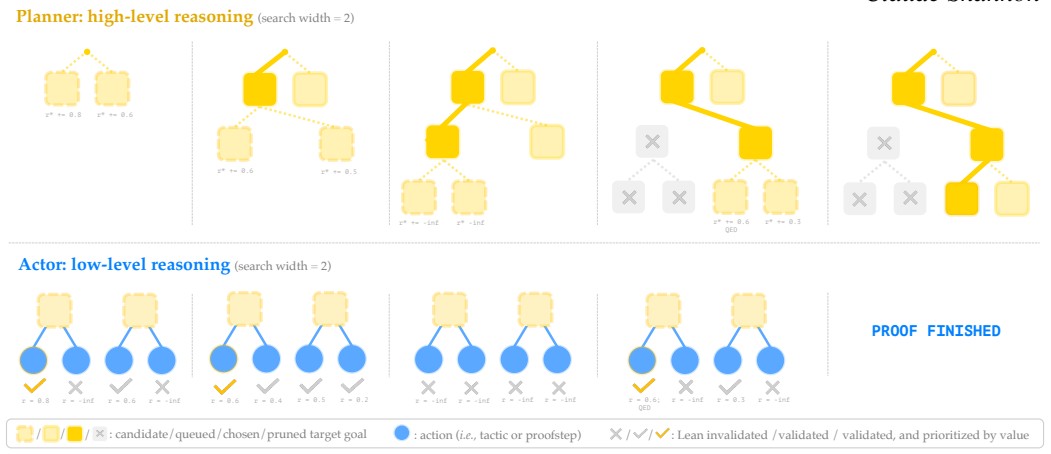

Figure 1: An illustrative example of Algorithm 1 and 2 on decomposing and search of RiR.

[*]*Part of this work was done at Eigent AI. Correspondence to: ziyuye@uchicago.edu, acgbull@gmail.com.
MATH-AI @ NeurIPS'24. All experiments and processing was conducted in Eigent AI and Caltech.*

# 1    Introduction

The main question we aim to address in this work is: what is an effective learning mechanism for language models to solve complex reasoning problems, such as mathematical theorem proving?

Recent progress in language models have shown promises in generating intermediate steps by next-token prediction [Wei et al., 2022], yet the performance often deteriorates when facing long trajectories or vast spaces. This challenge is particularly evident in automated theorem proving, a task that has been at the core of artificial intelligence research since the field's early days [Simon, 1969]. The process of crafting a proof is a classical example of reasoning [Wang, 1961]: just as a learning agent needs generalize from a limited set of examples to the broader set of all possible worlds, a prover must navigate from a given set of known theorems to the vast space of provable statements. An effective strategy from human mathematicians is the decomposition of problems with a sequence of target goals. This provides a more informative direction for subsequent reasoning steps, potentially reducing the effective search space.

We hereby introduce **Reasoning in Reasoning (RiR)**, a fundamental hierarchical framework unifying *decomposing* and *search*. In the context of theorem proving, our framework consists of an offline co-training stage followed by an online goal-driven bi-level planning stage. Our contributions are:

- **Framework**: We develop Reasoning in Reasoning (RiR), a new and general reasoning framework, that is practically implemented with goal-driven hierarchical learning via a planner-actor game for neural theorem proving.

- **Experiments**: We show that RiR achieves both state-of-the-art performance and efficiency on popular benchmarks of LeanDojo [Yang et al., 2023] and miniF2F [Zheng et al., 2021].

# 2    Preliminaries: Classical Neural Theorem Proving with Language Models

We here introduce classical methods. The glossary used throughout the paper is in Appendix A.

**Setup.**    We frame formal theorem proving as a Markov Decision Process. Starting with a to-prove statement $\mathbf{q}$ whose initial state is $\mathbf{s}_0$, we sequentially apply tactics $\mathbf{y}_t$ to prove it. Each tactic applied will make the current state $\mathbf{s}_t$ transit to the next state $\boldsymbol{s}_{t+1}$. Each state is associated with a scalar reward, $r(\mathbf{s}_t)$, provided by the environment. Below we show an example in Lean4.

```
theorem (p q: Prop): p v q → q v p := by   -- goal s₀: (p q: Prop) p v q → q v p
    intro h                                 -- goal s₁: (p q: Prop)(h: p v q) → q v p
    cases h with                            -- goal s₂: (p q: Prop)(hp: p) → q v p
     inl hp => apply Or.inr; exact hp       -- goal s₃: (p q: Prop)(hq: q) → q v p
     inr hq => apply Or.inl; exact hq       -- goal s₄: None
```

**Neural theorem proving.**    A neural network parameterized by $\boldsymbol{\theta}$ can act as a policy that samples single tactic $\mathbf{y}_{t+1} \sim \pi_{\boldsymbol{\theta}}(\cdot \mid \mathbf{s}_t)$ at step $t$. The objective is to find the optimal trajectory which leads to `solved` for each statement $\mathbf{q}$, that is to find a sequence of tactics $\mathbf{y}_1, \ldots, \mathbf{y}_T$ such that:

$$\mathbf{s}_0 \xrightarrow{\mathbf{y}_1} \mathbf{s}_1 \xrightarrow{\mathbf{y}_2} \mathbf{s}_2 \xrightarrow{\mathbf{y}_3} \ldots \xrightarrow{\mathbf{y}_T} \mathbf{s}_T.$$

The problem of automated theorem proving is often solved via a two-stage framework as follows.

**Stage 1: offline learning for proofstep generation.**    Classical approaches [Han et al., 2021, Welleck et al., 2022, Yang et al., 2023, Li et al., 2024] fine-tune a model $p_{\boldsymbol{\theta}}(\mathbf{y}^{\star} \mid \mathbf{s})$ to sample the next proofstep $\mathbf{y}$ conditional **only on current goal $\mathbf{s}$**. The classical prompt format is:

> **Input:**    {$current_goal s}    > **Output:**    {$proofstep y⋆}

**Stage 2: online search for complete proof.**    Classically, given a statement $\boldsymbol{q}$, a full proof $\bar{\mathbf{y}}_{1:T}$ is found by constructing a tree [Yang et al., 2023, Li et al., 2024] with **only low-level tactic search**. A common choice is *best-first search*, where there is a priority queue $\mathcal{Q}$ of partial proofs, ordered by some value function $v(\cdot)$. At step $t$, we pop one partial proof $\bar{\mathbf{y}}_{1:t}$ (each associated with its current state $\boldsymbol{s}_t$) with the highest value. We then expand $\bar{\mathbf{y}}_{1:t}$ by generating $M$ candidate proofsteps, and each resulting partial proof $\bar{\mathbf{y}}_{1:t+1} \in \mathcal{S}_{t+1}(\bar{\mathbf{y}}_{1:t})$ is inserted into the queue $\mathcal{Q}_{\bar{\mathbf{y}}}$ prioritized by the value. The search continues until a full proof $\bar{\mathbf{y}}_{1:T}$ is found, or termination criteria is reached.

## 3 Method

### 3.1 Offline Learning Stage: Goal-Driven Co-Training

Unlike classical approaches which learn to minimize the loss with regard to the conditional distribution $p(\mathbf{y}^\star \mid \mathbf{s})$, we propose to learn the joint distribution $p^\star(\mathbf{s}_{t+1}^\star, \mathbf{y}_{t+1}^\star \mid \mathbf{s}_t)$, where $\mathbf{s}_{t+1}^\star$ is the target goal state achieved by applying $\mathbf{y}_{t+1}^\star$. Our strategy is simple: we **co-train** a goal predictor model $p(\mathbf{s}^\star \mid \mathbf{s})$ and a goal-driven tactic predictor model $p(\mathbf{y}^\star \mid \mathbf{s}, \mathbf{s}^\star)$, with the co-training loss below:

$$\mathcal{L}_{\text{co}}(\boldsymbol{\theta}) = -\frac{1}{N} \underbrace{\sum_{(\mathbf{s}, \mathbf{y}^\star, \mathbf{s}^\star) \sim \mathcal{D}^{\text{train}}}}_{\text{triplet set}} \left[ \underbrace{\log p_{\boldsymbol{\theta}}(\mathbf{s}^\star \mid \mathbf{s})}_{\text{goal planner}} + \underbrace{\log p_{\boldsymbol{\theta}}(\mathbf{y}^\star \mid \mathbf{s}, \mathbf{s}^\star)}_{\text{goal-driven actor}} \right]. \quad (1)$$

We use the following input-output prompt format in training for the theorem proving task:

**Planner** (Target Goal Generation):

```
> Input:   [CURRENT GOAL] {$current_goal s} [TARGET GOAL]
> Output:  {$target_goal s*}
```

**Actor** (Goal-Driven Tactic Generation):

```
> Input:   [CURRENT GOAL] {$current_goal s} [TARGET GOAL] {$target_goal s*}
           [PROOFSTEP]
> Output:  {$tactic y*}
```

By decomposing the decision making process into goal state generation and goal-driven proofstep generation, RiR naturally captures the hierarchical structure of the reasoning.

### 3.2 Online Planning Stage: Goal-Driven Hierarchical Search

Algorithm 1 is a general design for RiR during the planning phase, where we can plug in various practical tree search policies. The high-level search explores promising target goals, while the low-level search finds the tactics to achieve each target goal, similar to the classical leader-follower game. A key feature is the joint update of both trees. An illustrative example is in Fig 1, and a concrete algorithm with best-first search (BFS) that we currently deploy for experiments is in Appendix D.

---

**Algorithm 1 RiR** – *A Unified Reasoning Mechanism with Decomposing and Search*

    **Input:** problem statement $\boldsymbol{q}$, a language model w/ parameter $\boldsymbol{\theta}$

1: $\overline{\texttt{tree}} \leftarrow \overline{\texttt{Tree}}(\boldsymbol{\theta}, \boldsymbol{q})$
2: **repeat**
3:     $\boldsymbol{s}_l^\star \leftarrow \overline{\texttt{tree}}.\texttt{policy}()$                         ▷   /* planner */
4:     $\underline{\texttt{tree}} \leftarrow \underline{\texttt{Tree}}(\boldsymbol{\theta}, \boldsymbol{s}_l^\star)$
5:     **repeat**
6:        $\boldsymbol{y}_{t_l} \leftarrow \underline{\texttt{tree}}.\texttt{policy}()$            ▷   /* goal-driven actor */
7:     **until** STOP_LOW
8:     $\{\overline{\texttt{tree}}, \underline{\texttt{tree}}\}.\texttt{update}()$           ▷   /* joint update */
9: **until** STOP_HIGH
10: **return** $\underline{\texttt{tree}}.\texttt{solution}$

---

## 4 Experiments

**Setups: Datasets and Models.** We use the random split of LeanDojo Benchmark 4 [Yang et al., 2023] as the training dataset. We use BYT5-0.3B [Xue et al., 2021] as our base model, which is a pretrained byte-level encoder-decoder Transformer model, and was adopted in Yang et al. [2023] with the state-of-the-art performance in theorem proving. We refer to this trained checkpoint of Reprover (w/o retrieval) as our baseline, and evaluate it *with the same setting* as RiR. We train the above model for 500K steps, with the learning rate as $5.0 \times 10^{-4}$ and batch size as 8. For evaluation, we use both LeanDojo Benchmark 4 and miniF2F [Zheng et al., 2021]. We use the *Pass@1* metric with 10-min timeout limit for evaluation. The search width for the high-level and the low-level is 5 and 64.

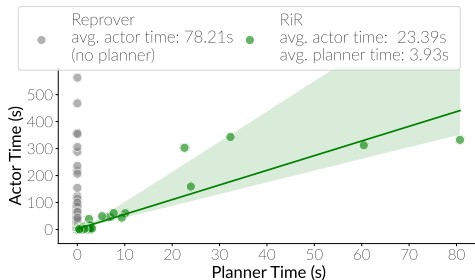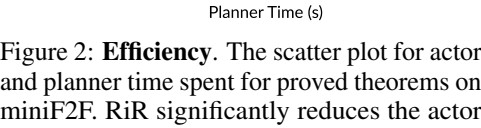

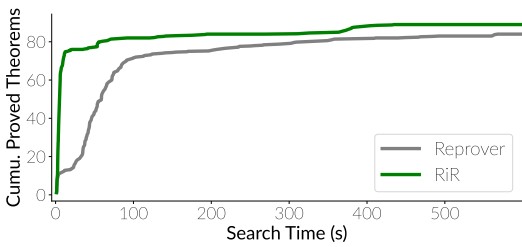

Figure 2: **Efficiency**. The scatter plot for actor and planner time spent for proved theorems on miniF2F. RiR significantly reduces the actor time via the goal guidance from the planner.

Figure 3: **Efficiency**. The CDF plot for search time spent for proved theorems on miniF2F Benchmark. RiR is significantly faster (nearly **3x**) than the existing state-of-the-art baseline.

**Results: Performance Gain.**    We present the performance comparison of RiR with existing baselines in Table 1. RiR also proved 1 more AIME and 2 more AMC problems compared to the current state-of-the-art Reprover [Yang et al., 2023].

| Dataset ($\rightarrow$) | miniF2F-test[2] | LeanDojo-test |
|---|---|---|
| **Method ($\downarrow$) / Model ($\rightarrow$)** | BYT5-0.3B | BYT5-0.3B |
| Reprover (BFS) | 34.43% | 50.16% |
| RiR (BFS) | **36.89%** | **53.73%** |

Table 1: **Performance.** *Pass@1* rate on LeanDojo and miniF2F.

**Results: Efficiency Gain.**    RiR is significantly faster in searching for the optimal reasoning trajectories via a more compact and information-directed search space with the goal-driven planner, as illustrated in Figure 3 on miniF2F benchmark. RiR is more time efficient in the sense that we achieve better results in small computational budget. Specifically, as shown in Figure 2, while the classical Reprover has an average actor time (*i.e.*, time spent for low-level proofstep search) of 78.21s, RiR reduces this to only **23.39**s, with additional 3.93s for planner time (*i.e.*, time spent for high-level goal search) on average, setting the new efficiency benchmark for neural theorem proving.

**Remarks.**    We present logs showing how RiR found hard proofs fast while classical approaches fail in Appendix F; take `Finset.union_subset_left` for example, while the classical method expanded more than 8914 nodes yet still failed after 10 minutes, RiR proved the theorem within 5 seconds and only searched 1 node. We believe the significant improvement in efficiency and effectiveness comes from RiR's ability to generalize better and to explore better in the more compact and informative search space, empirically supporting the Conjecture 2 and 3 in Appendix B.

## 5    Conclusions

We have developed Reasoning in Reasoning (RiR), an easy-to-implement framework unifying reasoning by search and reasoning by decomposing for language models. In the domain of automated theorem proving, RiR is practically implemented with goal-driven offline pretraining and hierarchical online planning, where reasoning takes place in different semantic levels. We explore RiR with initial information-theoretical analysis, discussing the Co-Training Advantage Conjecture and the Hierarchical Planning Advantage Conjecture in Appendix B, and present detailed discussions in related works and limitations in Appendix C and E. We hope RiR can shed light on the fundamental way for reasoning with language models.

---

[2]Additional data points on miniF2F-test for readers' reference:

- Azerbayev et al. [2023] LLEMMA-7B achieves 26.2%;
- Welleck and Saha [2023] LLMSTEP-1.8B achieves 27.9%;
- Polu and Sutskever [2020] GPT-$f$ achieves 36.6%.

## Acknowledgements

This research was also supported in part through grants from the U.S. Department of Energy under Grant No. DOE DE-EE0009505, the National Science Foundation under Grant No. IIS 2332475, with additional acknowledgment to the University of Chicago's Research Computing Center. All experimentation and processing were conducted solely on Eigent AI and Caltech servers.

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

# Appendix

## A  Glossary

| | |
|---|---|
| *theorem statement* ($\mathbf{q}$) | a mathematical statement. |
| *goal* ($\mathbf{s}$) | a statement in the context of a proofsearch, denoted as $\mathbf{s}$. |
| *state* | a representation containing contexts (*e.g.*, hypotheses) and goals for the proof; for simplicity, we also use this term interchangeably with *goal*. |
| *proofstep / tactic* ($\mathbf{y}$) | a reasoning step that uses established assumptions etc to achieve the goal. |
| | |
| *reasoning* | the process of deriving intermediate steps to solve a problem. |
| *planning* | a sub-type of reasoning on deriving high-level goals that trigger low-level steps. |
| *low-level search* | the sampling and pruning for *proofsteps*. |
| *high-level search* | the sampling and pruning for *goals*, see Section 3 for details. |

## B  Theoretical Conjectures: An Information Gain Perspective

The simple insight is that the new mechanism of RiR *increases information learned from environments*, improving both generalization for reasoning step learning and exploration for reasoning path planning.

### B.1  Intuition

To recap, we propose a hierarchical approach for learning in theorem proving:

1. Planner step: predicting the target state via $p(\mathbf{s}_{t+1}^\star \mid \mathbf{s}_t)$.
2. Actor step: predicting the proofstep via $p(\mathbf{y}_t^\star \mid \mathbf{s}_t, \mathbf{s}_{t+1}^\star)$.

This contrasts with the traditional approach of solely predicting $p(\mathbf{y}_t^\star \mid \mathbf{s}_t)$.

Let's think in an information-theoretic way: $\mathbf{s}_{t+1}$ **acts as an information bottleneck** [Shwartz-Ziv and Tishby, 2017], by *abstracting* different possible proofsteps or sequences of proofsteps $\mathbf{y}_t$ into a single, more compact representation. Consider a simplified example below:

```
-- goal sₜ   =   3 * (2 + 1)  = 9
-- goal sₜ₊₁  =   9 = 9
```

There exist multiple different proofsteps to reach $\mathbf{s}_{t+1}$ from $\mathbf{s}_t$, for instance:

- `ring` – algebraic normalization.
- `norm_num` – direct numeric evaluation.
- `simp; rfl` – simplification followed by reflexivity.
- `calc ⋯ (omitted)` – step-by-step calculation.

It is important that in this way, $\mathbf{s}_{t+1}$ could generalize beyond our current setting (*i.e.*, the next formal goal in Lean4). Essentially, it can be any **abstraction of the formal proofsteps**, for example:

- A high-level thought expressed in informal natural language.
- A discrete code representing a proof strategy.
- A latent vector in a learned numeric representation space.

Intuitively, the abstraction brought by $\mathbf{s}_{t+1}^\star$ helps capture essential proof structure and compress away irrelevant details, which can also be considered as a way to reduce estimation errors [Jiang, 2018].

Information never hurts [Cover, 1999] – there is $H(\mathbf{y}_t^\star \mid \mathbf{s}_t, \mathbf{s}_{t+1}^\star) \leq H(\mathbf{y}_t^\star \mid \mathbf{s}_t)$, *i.e.*, knowing $\mathbf{s}_{t+1}^\star$ helps reduces uncertainty about $\mathbf{y}_t^\star$, providing a more focused direction for action search. In the theorem-proving scenario, one may assume a lower bound on this additional knowledge. We now present preliminary theoretical conjectures as follows.

## B.2 Generalization Guarantee for Goal-Driven Policy Co-Training

**Assumption 1** *The conditional mutual information between the optimal action $\mathbf{y}^\star$ and the optimal target goal $\mathbf{s}^\star$, given the current state $\mathbf{s}$, is bounded by a constant $\gamma_I > 0$:*

$$I(\mathbf{y}^\star; \mathbf{s}^\star | \mathbf{s}) \geq \gamma_I. \tag{2}$$

**Assumption 2** *Let $p^\star(\mathbf{s}, \mathbf{s}^\star, \mathbf{y}^\star)$ be the true joint distribution over triplets $\{(\boldsymbol{s}_i, \boldsymbol{y}_i^\star, \boldsymbol{s}_i')\}_{i=1}^N$. Let $p_{\boldsymbol{\theta}_c}(\mathbf{y}^\star \mid \mathbf{s})$ and $p_{\boldsymbol{\theta}_{co}}(\mathbf{y}^\star \mid \mathbf{s})$ be the learned distributions for the classical and the co-training approach from minimizing the empirical loss $\mathcal{L}_c(\boldsymbol{\theta})$ of classical method and $\mathcal{L}_{co}(\boldsymbol{\theta})$ in Eq. 1. We assume:*

1. *The hypothesis classes $\Theta_c$ and $\Theta_{co}$ have VC dimensions $d_c$ and $d_{co}$, and are such that:*

$$\mathcal{L}_c(\vartheta_c^*) \leq \inf_{\boldsymbol{\theta} \in \Theta_c} \mathcal{L}_c(\boldsymbol{\theta}) + O\left(\sqrt{\frac{d_c + \log(1/\delta)}{N}}\right),$$

$$\mathcal{L}_{co}(\vartheta_{co}^*) \leq \inf_{\boldsymbol{\theta} \in \Theta_{co}} \mathcal{L}_{co}(\boldsymbol{\theta}) + O\left(\sqrt{\frac{d_{co} + \log(1/\delta)}{N}}\right),$$

*with probability at least $1 - \delta$ over the choice of the training set, where $\vartheta_c^* = \arg\min_{\boldsymbol{\theta} \in \Theta_c} \mathcal{L}_c(\boldsymbol{\theta})$ and $\vartheta_{co}^* = \arg\min_{\boldsymbol{\theta} \in \Theta_{co}} \mathcal{L}_{co}(\boldsymbol{\theta})$.*

2. *The number of training examples $N$ is sufficiently large such that $N \geq \frac{32(d_{co} + \log(1/\delta))}{\gamma_I}$.*

**Conjecture 1 (Loss Decomposition with Information Gain)** *Let $\mathcal{L}_{co}(\vartheta_{co}^*)$ be the optimal co-training loss and $\mathcal{L}_c(\vartheta_c^*)$ be the optimal classical loss. Suppose the conditional mutual information satisfies $I(\mathbf{y}^\star; \mathbf{s}^\star \mid \mathbf{s}) \geq \gamma_I$ for some constant $\gamma_I > 0$. Then, there exists a constant $C > 0$ such that:*

$$\mathcal{L}_{co}(\vartheta_{co}^*) \leq \mathcal{L}_c(\vartheta_c^*) + C\gamma_I.$$

**Conjecture 2 (Co-Training Advantage)** *By Assumption 1 and 2, with probability at least $1 - 2\delta$ over the choice of the training set, the following inequality holds:*

$$\mathbb{E}_{p^\star(\mathbf{s})}\left[\|p^\star(\mathbf{y}^\star \mid \mathbf{s}) - p_{\boldsymbol{\theta}_c}(\mathbf{y}^\star \mid \mathbf{s})\|_{TV}\right] \geq \mathbb{E}_{p^\star(\mathbf{s})}\left[\|p^\star(\mathbf{y}^\star \mid \mathbf{s}) - p_{\boldsymbol{\theta}_{co}}(\mathbf{y}^\star \mid \mathbf{s})\|_{TV}\right] + |f(C\gamma_I)|,$$

*where $f(\cdot)$ is assumed to be monotonic.*

## B.3 Efficiency Guarantee for Goal-Driven Hierarchical Planning

In our hierarchical approach for theorem proving, we introduce a target goal space $\tilde{\mathcal{S}} = \mathcal{S}$. At step $t$, given the current state $\mathbf{s}_t$, we first search for target goals given the current goal; next, conditional on the chosen target goals, we search for tactics, and apply the chosen tactic to transit to new states; the process repeats until termination. In contrast, the classical single-level planning approach only samples low-level tactics without any high-level guidance; we refer to this as the flat planning:

- (Classical) **Flat planning**: we have a policy $\pi_f : \mathcal{S} \to \mathcal{A}$ that maps states to actions.
- (RiR) **Hierarchical planning**: we have:
  - A *high-level planner* policy $\pi_h : \mathcal{S} \to \tilde{\mathcal{S}}$, that maps goals to target goals.
  - A *low-level actor* policy $\pi_l : \mathcal{S} \times \tilde{\mathcal{S}} \to \mathcal{A}$, that maps goals and target goals to actions.

The simple intuition is that the introduction of the goal state creates a partitioning over the raw action space, reducing the search space and making the search more efficient.

**Conjecture 3 (Hierarchical Planning Advantage)** *Consider a hierarchical planning approach with a high-level policy $\pi_h$ and a low-level policy $\pi_l$, and a flat planning approach with a policy $\pi_f$. Let $N_h(\epsilon)$ and $N_f(\epsilon)$ be the number of node expansions required by the hierarchical and flat planning approaches to find an $\epsilon$-optimal solution w.p. at least $1 - \delta$. Under mild assumptions, there exist constants $c_1, c_2, \gamma > 0$ such that:*

$$\mathbb{E}\left[N_h(\epsilon)\right] \leq c_1 e^{-I(\mathcal{A};\tilde{\mathcal{S}}|\mathcal{S})} \cdot \log\left(\frac{1}{\delta}\right) \cdot \left(\mathbb{E}\left[N_f(\epsilon)\right]\right)^{\gamma} + c_2 \cdot \psi(\epsilon_h, \epsilon_l), \tag{3}$$

*where $\gamma_I$ is the conditional mutual information between the optimal action and the optimal target goal, and $\epsilon_h$ and $\epsilon_l$ are the $\epsilon$-optimality gaps of the learned high-level and low-level policies, respectively.*

In essence, RiR is helpful when target goals effectively decompose the problem into smaller subproblems while preserving the essential information about the optimal solution. Intuitively, if the target goals selected by the high-level policy provide useful information towards the optimal actions, the low-level policy can focus on a smaller set of relevant actions, leading to more efficient search.

## C   Related Works

**Reasoning with language models.**   In language modeling, reasoning typically refers to generating intermediate steps within the language space to reach a final solution to a problem [Wei et al., 2022]. Solving complex or novel reasoning problems remains as an open challenge. One promising direction is **reasoning by searching**, *e.g.*, expanding the reasoning space by tree search for intermediate steps [Yao et al., 2024, Feng et al., 2023, Liu et al., 2023a, Yuan et al., 2024]. Another research direction is **reasoning by decomposition**, *i.e.*, generating higher-level goals that trigger a single or a sequence of lower-level steps [Zhou et al., 2022, Liu et al., 2023b, Zheng et al., 2023, Liang et al., 2024, Dalal et al., 2024, Huang et al., 2022, Hu et al., 2024]. The most similar line of literature to ours is subgoal search [Wilkins, 1980, Czechowski et al., 2021, Zawalski et al., 2022, Parascandolo et al., 2020, Paul et al., 2019], while we put specific focus on the theorem proving benchmarks, and present initial theoretical conjectures, and formally unifies *search* and *decomposing* in a hierarchical framework for large language model training and inference.

**Automatic Theorem Proving with language models.**   As a representative reasoning task, automatic theorem proving (ATP) is often characterized as a *tree search* problem, *i.e.*, constructing a (tactic-based) proof tree and traversing it to find the correct proof [Li et al., 2024]. In the context of language modeling, *proofstep generation* forms the edges of the proof tree; the common standard in prior works is to generate single proof steps with the input format similar to `[GOAL]${goal}[PROOFSTEP]`, *i.e.*, conditional on the current goal, generating the next tactic [Polu and Sutskever, 2020, Yang et al., 2023, Azerbayev et al., 2023, Lample et al., 2022]. For the proof search stage, while people have been using simple heuristics like breadth-first search [Bansal et al., 2019], or MCTS-like search guided by learned value functions [Lample et al., 2022, Polu et al., 2022], designing better search algorithms remains an active area [Li et al., 2024]. The key challenge is that the tactic-based proof space is combinatorially large. Distinguished from prior works , RiR introduces the goal-driven co-training for proofstep generation with a bi-level search framework for generalization and efficiency advantage.

**Hierarchical and goal-conditioned RL.**   Planning and learning is hard when the decision-making space scales up [Bakker et al., 2004]. Hierarchical RL intends to address this issue by learning a hierarchy of policies operating on different levels of abstraction (*e.g.*, subgoals over the state space). This mitigates the scaling issues by improving exploration for the environment [Ghosh et al., 2020, Chitnis et al., 2022, Kumar et al., 2023, Silver et al., 2023, Le et al., 2018]. There is another line of research termed as goal-conditioned RL [Ghosh et al., 2019, Wang et al., 2023, Ghugare et al., 2024], which trains offline RL policy in a supervised manner conditioning on goal or return. Unlike most prior works that rely on a predefined goal structure, we train models to *learn to generate goals* in the language space, and refine the goal planning via low-level tree search and joint update.

## D   A Practical Implementation of RiR with Best-First Search

Here, we propose a bi-level best-first search algorithm which maintains a priority queue of trajectories, where the priority of a trajectory is determined by its *joint* negative log-likelihood, defined as:

$$-\log p(\tau) = -\sum_{i=1}^{t} \log p(\boldsymbol{y}_{i+1}, \boldsymbol{s}_{i+1}^{\star}|\boldsymbol{s}_i) = -\sum_{i=1}^{t}\left(\log p(\boldsymbol{s}_i^{\star}|\boldsymbol{s}_{i-1}) + \log p(\boldsymbol{y}_{i+1}|\boldsymbol{s}_{i+1}^{\star}, \boldsymbol{s}_i)\right).$$

At each iteration, the algorithm pops the highest-priority trajectory. It performs high-level search to sample target goals, and low-level search to sample tactics conditioned on both the target and the

current goal. By prioritizing trajectories with policy heuristics, RiR efficiently explores the most promising reasoning paths, using the learned model to guide both goal planning and tactic generation.

---

**Algorithm 2** RiR – Best-First Search

**Input:** problem statement $q$, a language model with parameter $\theta$

1: $\mathcal{Q} \leftarrow \text{QUEUE}(q)$
2: **while** $\mathcal{Q} \neq \emptyset$ and not $\text{BUDGETEXHAUSTED}()$ **do**
3: $\quad \tau = (s_0, (\hat{s}_1^\star, \hat{y}_1), \ldots, s_{t-1}, (\hat{s}_t^\star, \hat{y}_t), s_t) \leftarrow \mathcal{Q}.\text{POP}()$

4: $\quad$ **if** $s_t$ is $\text{PROOFFINISHED}$ **then return** $\tau$
5: $\quad$ **end if**

6: $\quad \mathcal{G}_t \leftarrow \text{SAMPLETARGETGOALS}(s_t, \theta)$ $\qquad\qquad\qquad \triangleright \quad /^\star$ high-level search $^\star/$
7: $\quad$ **for** $\hat{s}_t^{\star(i)} \in \mathcal{G}_t$ **do**
8: $\quad\quad \mathcal{Y}_t^{(i)} \leftarrow \text{SAMPLETACTICS}(\hat{s}_{t+1}^{\star(i)}, s_t, \theta)$ $\qquad\qquad \triangleright \quad /^\star$ low-level search $^\star/$
9: $\quad\quad$ **for** $\hat{y}_{t+1}^{(i,j)} \in \mathcal{Y}_t^{(i)}$ **do**
10: $\quad\quad\quad s_{t+1}^{(i,j)} \leftarrow \text{APPLYTACTIC}(\hat{y}_{t+1}^{(i,j)}, s_t)$
11: $\quad\quad\quad \tau' \leftarrow (s_0, (\hat{s}_1^\star, \hat{y}_1), \ldots, s_t, (\hat{s}_{t+1}^\star, \hat{y}_{t+1}), s_{t+1}^{(i,j)})$
12: $\quad\quad\quad \mathcal{Q}.\text{PUSH}(\tau', -\log p(\tau'))$ $\qquad\qquad\qquad \triangleright \quad /^\star$ joint update $^\star/$
13: $\quad\quad$ **end for**
14: $\quad$ **end for**

15: **end while**
16: **return** $\text{FAILURE}$

---

Note that the Best-First Search here can be easily switched to other search algorithms, *e.g.*, Monte Carlo Tree Search [Coulom, 2006] or scalable RL finetuning [Fickinger et al., 2021], which we encourage the community to explore in more depth.

## E  Limitations and Future Steps

While we have shown the effectiveness of RiR on neural theorem proving benchmarks, there are a lot more to be built upon our framework. Future directions may include: (i) incorporating dedicated reward models in the planning phase (rather than using the likehood heuristics); (ii) adding post-training during planning using techniques like contrastive preference learning [Hejna et al., 2023], to further tune the model with pairs of successful and failed reasoning trajectories for self-improvement [Hosseini et al., 2024]; (iii) integrating language feedback [Cheng et al., 2023], contextual information [Welleck and Saha, 2023], and other broader goals for co-training and planning; (v) investigating in-context learning alternatives for co-traning to apply RiR in black-box models; (vii) adding goal rollout and lookahead to further improve RiR's performance and efficiency.

## F  Detailed Experimental Results and Logs

We are open-sourcing all our codes, training scripts, evaluation logs, and checkpoints at this link:

- github.com/ziyu-deep/reasoning-in-reasoning.

For evaluation on LeanDojo, we use:

- Repository: https://github.com/leanprover-community/mathlib4.
- Commit: fe4454af900584467d21f4fd4fe951d29d9332a7.

For evaluation on miniF2F, we use:

- Repository: https://github.com/yangky11/miniF2F-lean4.
- Commit: 9e445f5435407f014b88b44a98436d50dd7abd00.

We hereby present some example proofs from logs, showing how RiR succeeded with significantly fewer nodes to search. More examples can be found in our released repository.

```
Example 0:  Proof Found by RiR

Theorem:
  File Path: Mathlib/Order/ConditionallyCompleteLattice/Basic.lean
  Full Name: OrderIso.map_ciSup

Status: Status.PROVED

Proof:
  simp [iSup, hf]
  rw [e.map_csSup']
  swap
  assumption'
  apply Set.range_nonempty
  rw [← Set.range_comp]
  rfl

Search Statistics:
  Planner Time: 150.2634212092962
  Actor Time: 315.0649007729953
  Environment Time: 38.92193151102401
  Total Time: 505.9431369260419
  Total Nodes: 2207
  Searched Nodes: 37
```

```
Example 0:  Failure by Reprover (w/o retrieval)

Theorem:
  File Path: Mathlib/Order/ConditionallyCompleteLattice/Basic.lean
  Full Name: OrderIso.map_ciSup

Status: Status.OPEN

Proof: None

Search Statistics:
  Actor Time: 512.3867035790754
  Environment Time: 89.58101247090963
  Total Time: 602.1384408420126
  Total Nodes: 4082
  Searched Nodes: 160
```

```
Example 1:  Proof Found by RiR

Theorem:
  File Path: Mathlib/Data/Finset/Basic.lean
  Full Name: Finset.union_subset_left

Status: Status.PROVED

Proof:
  exact Finset.Subset.trans (Finset.subset_union_left s t) h

Search Statistics:
  Planner Time: 1.3937767379684374
  Actor Time: 3.304290219093673
  Environment Time: 0.07375576300546527
  Total Time: 4.774586059036665
  Total Nodes: 7
  Searched Nodes: 1
```

```
Example 1:  Failure by Reprover (w/o retrieval)

Theorem:
  File Path: Mathlib/Data/Finset/Basic.lean
  Full Name: Finset.union_subset_left

Status: Status.OPEN

Proof: None

Search Statistics:
  Actor Time: 491.1531239761098
  Environment Time: 110.1171304465338
  Total Time: 601.520013278001
  Total Nodes: 8914
  Searched Nodes: 233
```

```
Example 2:  Proof Found by RiR

Theorem:
  File Path: Mathlib/Data/Nat/PrimeFin.lean
  Full Name: Nat.Prime.primeFactors

Status: Status.PROVED

Proof:
  ext
  simp [hp.ne_zero]
  simp [hp, Nat.dvd_prime hp]
  aesop

Search Statistics:
  Planner Time: 150.2634212092962
  Actor Time: 315.0649007729953
  Environment Time: 38.92193151102401
  Total Time: 505.9431369260419
  Total Nodes: 2207
  Searched Nodes: 37
```

```
Example 2:  Failure by Reprover (w/o retrieval)

Theorem:
  File Path: Mathlib/Data/Nat/PrimeFin.lean
  Full Name: Nat.Prime.primeFactors

Status: Status.OPEN

Proof: None

Search Statistics:
  Actor Time: 474.4240076234564
  Environment Time: 127.5987611755263
  Total Time: 602.1851601980161
  Total Nodes: 4231
  Searched Nodes: 133
```

```
Example 3:  Proof Found by RiR

Theorem:
  File Path: Mathlib/Order/SuccPred/Basic.lean
  Full Name: exists_succ_iterate_or

Status: Status.PROVED

Proof:
  obtain h | h := le_total a b
  exacts [Or.inl (IsSuccArchimedean.exists_succ_iterate_of_le h),
  Or.inr (IsSuccArchimedean.exists_succ_iterate_of_le h)]

Search Statistics:
  Planner Time: 15.921687303110957
  Actor Time: 44.464585242792964
  Environment Time: 8.429574175737798
  Total Time: 68.86368872597814
  Total Nodes: 377
  Searched Nodes: 3
```

```
Example 3:  Failure by Reprover (w/o retrieval)

Theorem:
  File Path: Mathlib/Order/SuccPred/Basic.lean
  Full Name: exists_succ_iterate_or

Status: Status.OPEN

Proof: None

Search Statistics:
  Actor Time: 519.0408471203409
  Environment Time: 86.30267171841115
  Total Time: 605.4483464460354
  Total Nodes: 2819
  Searched Nodes: 95
```

