# OpenReview forum: "Reasoning in Reasoning: A Hierarchical Framework for Better and Faster Neural Theorem Proving"
_NeurIPS.cc/2024/Workshop/MATH-AI — MATH-AI 24_

### Official Review · Reviewer_JTye · 2024-10-01
**Review for "Reasoning in Reasoning: A Hierarchical Framework for (Better and Faster) Neural Theorem Proving"**

**Rating:** 9
**Confidence:** 4

**Review:**

This paper introduces the “Reasoning in Reasoning” (RiR) framework for improving neural theorem proving through hierarchical problem decomposition and goal-driven search. RiR is designed to tackle complex reasoning tasks by splitting them into manageable sub-goals, improving both generalization and efficiency. The approach leverages a planner-actor game structure and combines offline co-training with online planning stages, evaluated through benchmarks like LeanDojo and miniF2F. Overall, I think it’s a great paper and I believe the contributions in this paper will benefit the audience and readership within this field.
Some suggestions on improvements:
(1)	While RiR performs well in the specific context of neural theorem proving, the framework's generalizability to other types of reasoning tasks is unclear. Can the authors discuss the wider applicability of this framework in introductions and conclusions?
(2)	The authors do not provide concrete experiments to test the Co-Training Advantage Conjecture and Hierarchical Planning Advantage Conjecture, leaving the theoretical contributions somewhat speculative. Empirical validation of these conjectures would enhance the credibility of the information-theoretic insights. I suggest authors also incorporate some empirical investigations on this part.
(3)	The paper highlights several success stories where RiR outperforms classical methods, but it does not offer a sufficient discussion of its failure cases. Understanding where and why RiR fails is crucial for improving the method and providing a more nuanced perspective on its limitations. I suggest authors include a thorough analysis of RiR’s failure cases to better understand its limitations. This could guide future improvements and offer a more balanced view of its performance.
(4)	While the paper compares RiR with the Reprover baseline, which uses a non-hierarchical approach, it does not compare RiR with other existing hierarchical or goal-conditioned reinforcement learning (RL) techniques. I suggest authors comparing RiR with other hierarchical or goal-conditioned reasoning and reinforcement learning methods.

---

### Official Review · Reviewer_rnmL · 2024-10-03
**The paper proposes a novel hierarchical framework, Reasoning in Reasoning (RiR), that combines goal-driven decomposition and efficient search strategies to tackle complex theorem proving.**

**Rating:** 7
**Confidence:** 3

**Review:**

**Summary:**
The paper proposes a novel hierarchical framework, Reasoning in Reasoning (RiR), that combines goal-driven decomposition and efficient search strategies to tackle complex theorem proving. Using neural theorem proving as a representative task, the approach achieves state-of-the-art performance and efficiency on benchmarks like LeanDojo and miniF2F.

**Pros:**
- **Simple Approach**: The hierarchical combination of decomposition and search effectively narrows down the search space, enhancing performance on challenging theorem-proving tasks.
- **Efficiency and Scalability**: RiR’s hierarchical structure reduces the search space and planning time, making it significantly faster than baselines.

**Cons:**
- **Over-reliance on Goal Selection**: The effectiveness of the RiR framework hinges on accurate goal prediction. Misestimating goals can lead to poor tactic generation and search inefficiencies.
- **Sparse Reward Signal**: The sparse reward setting in RiR may hinder learning in more nuanced settings where intermediate success is valuable.

**Questions:**
- How sensitive is RiR to the choice of goal representations? Would different abstractions lead to varying levels of performance and generalization?
- How does the complexity of hierarchical search compare to traditional tree search methods for more difficult proofs?

---

### Official Review · Reviewer_x25m · 2024-10-08
**A novel hierarchical framework for neural theorem proving**

**Rating:** 5
**Confidence:** 4

**Review:**

The paper introduces "Reasoning in Reasoning" (RiR), a novel hierarchical framework for neural theorem proving. The presentation, including figures and mathematical formulations, is well-executed. While the approach shows promise, further evaluation is needed to fully validate the authors' claims before acceptance.

## Pros:
1. The authors provide code, enhancing reproducibility and credibility.
2. This paper has relatively substantial mathematical proofs.
3. RiR demonstrates significantly faster performance and improved accuracy compared to existing baselines on the miniF2F benchmark.

## Cons:
### Evaluation:
The evaluation is limited to miniF2F-test and LeanDojo-test, using only the Pass@1 metric and best-first search. To strengthen their claims of being "better and faster," the authors should:
1. Include additional metrics (e.g., Pass@5, Pass@10) on a wider range of popular benchmarks.
2. Provide more comprehensive execution time and performance comparisons across various benchmarks and search methods.
3. Present more extensive experimental results under diverse settings to solidify their claims.
4. Release the time comparison result on LeanDojo-test.

### Writing:
In Figure 1, there appears to be an inconsistency in the high-level planning update. After the third low-level planning stop, the $S^{(1)}_3$ is correct, but the $S^{(2)}_3$ is chosen for the high-level planning update, which is confusing. And in the last stage, $S^{(1)}_2$ apears twice.

---

### Official Review · Reviewer_Tq6y · 2024-10-09
**Clear improvement + Theoretical analysis - Lacking novelty - Limited evaluation -> Marginal rejection**

**Rating:** 5
**Confidence:** 3

**Review:**

### Contributions

1. Clear improvement over the baseline.
2. Information-theoretic analysis for why the high-level planning helps.

### Weaknesses

1. The idea and setting lack novelty: The idea of high-level planning before low-level steps and corresponding practice of co-training on neural theorem proving tasks are explored by many previous works, such as Lean-STaR and DeepSeekProver.
2. The evaluation is limited: The upper limit of proof success rate is more important for the theorem proving task, while the efficiency is secondary. Specifically, the paper only evaluates the methods with metrics like pass@1 and time-relevant ones. For improvement, the paper should adopt more upper-limit-like metrics like pass@k with a large k.

---

### Decision · Program_Chairs · 2024-10-08

Accept